# Towards Intelligent Assessment in Personalized Physiotherapy with Computer Vision

**DOI:** 10.3390/s25113436

**Published:** 2025-05-29

**Authors:** Victor García, Olga C. Santos

**Affiliations:** PhyUM Reserch Center, Department of Artificial Intelligence, Computer Science School, UNED, 28040 Madrid, Spain; vgarcia569@alumno.uned.es

**Keywords:** YOLO Pose, pose estimation, optical sensors, computer vision, physical therapy assessment, semantic framework

## Abstract

**Highlights:**

This article explores the integration of computer vision in physiotherapy, evaluating the YOLO Pose algorithm for real-time assessment of human movement and proposing a semantic framework to translate motion data obtained from optical sensors, such as RGB cameras and other vision-based sensing technologies, which capture human movement data non-invasively, into clinically relevant parameters.

**What are the main findings?**
YOLO Pose can detect physiotherapy-relevant body keypoints with sufficient speed and efficiency for real-time applications.A semantic framework was developed to map pose estimation data into clinically interpretable parameters like strength, balance, and coordination.

**What is the implication of the main finding?**
Automated motion analysis can improve objectivity and consistency in physiotherapy assessments.The proposed approach reduces the time spent on manual evaluation and supports data-driven clinical decision making.

**Abstract:**

Effective physiotherapy requires accurate and personalized assessments of patient mobility, yet traditional methods can be time-consuming and subjective. This study explores the potential of open-source computer vision algorithms, specifically YOLO Pose, to support automated, vision-based analysis in physiotherapy settings using information collected from optical sensors such as cameras. By extracting skeletal data from video input, the system enables objective evaluation of patient movements and rehabilitation progress. The visual information is then analyzed to propose a semantic framework that facilitates a structured interpretation of clinical parameters. Preliminary results indicate that YOLO Pose provides reliable pose estimation, offering a solid foundation for future enhancements, such as the integration of natural language processing (NLP) to improve patient interaction through empathetic, AI-driven support.

## 1. Introduction

The accurate assessment of human mobility is a critical component in physiotherapy, particularly in cases involving musculoskeletal and neurological conditions. The global burden of disability is rising: musculoskeletal disorders alone affect over 1.7 billion people worldwide, and stroke remains a leading cause of long-term impairment [1,2]. At the same time, health systems face increasing pressure to deliver high-quality care with limited resources and an aging population [3]. These challenges have driven the development of tele-rehabilitation and eHealth solutions. However, many remote programs still rely on subjective patient self-reports or require specialized hardware that is costly and difficult to scale [4,5].

Traditionally, clinicians have relied on visual and manual evaluations—methods that, while widely accepted, are often time-consuming and subjective, leading to variability in diagnosis and treatment decisions [1,2,3]. This is particularly problematic in cases involving neurological impairments, where subtle variations in movement can signal significant dysfunction [2,3]. To address this, a wide array of clinical scales has been developed to assess key parameters such as range of motion, strength, coordination, balance, and pain. Tools like the National Institutes of Health Stroke Scale (NIHSS), Barthel Index, Modified Ashworth Scale (MAS), Berg Balance Scale, and Motor Index (MI) provide a standardized vocabulary for assessing functionality and guiding treatment. Conversions between qualitative and quantitative metrics—e.g., the Visual Analog Scale (VAS) or the Karnofsky Scale—further support clinical decision making.

However, the multiplicity of available tools and scales can complicate data management and hinder the integration of patient information. This highlights the need for a unified framework capable of translating diverse clinical data into abstract, high-level parameters—particularly when working with rich data sources such as those derived from computer vision and motion capture technologies that collect data from optical sensors. Implementing such a framework would simplify information management, enhance diagnostic consistency, and allow clinicians to dedicate more time to patient care by reducing administrative burdens.

In recent years, physiotherapy has undergone a transformation through the adoption of advanced technologies, particularly motion capture and movement analysis tools. These technologies allow for the acquisition of precise and objective kinematic data, offering a quantitative basis for evaluating motor function and neuromuscular dysfunctions [4,5,6]. Motion capture systems and inertial sensors have made it possible to visualize the dynamics of human movement in detail, significantly improving the reproducibility and accuracy of clinical assessments [3,7].

Despite their potential, a key challenge persists mapping raw motion data to clinically meaningful parameters. While scales such as the Berg Balance Scale, the Motor Index, or the NIH Stroke Scale (NIHSS) provide structured assessment criteria [8,9], they often rely on observation and patient interaction rather than direct sensor data input. Bridging this gap requires developing semantic models that translate quantitative measurements into clinically interpretable variables—a critical step for improving diagnostic accuracy and standardizing physiotherapeutic interventions [2,10,11].

In this context, computer vision techniques have emerged as a promising avenue. By capturing and analyzing visual information from standard video input obtained from optical sensors such as cameras, these technologies simulate human perception and provide real-time, non-invasive assessments of movement [12]. Among them, pose estimation algorithms stand out for their ability to identify and track body keypoints, facilitating detailed biomechanical evaluations [13,14]. Given the clinical importance of tracking individual body segments rather than treating the body as a single entity, this study proposes a segmentation strategy during data labeling that differentiates between the five limbs and the trunk. Such anatomical specificity is expected to enhance the precision of movement analysis and facilitate more clinically meaningful interpretations of collected data. The analysis of human movement is a complex task involving multiple joints and muscle groups, with applications in health, sports, and rehabilitation. Among various open-source solutions, the YOLO Pose algorithm [15,16] stands out for its efficiency and speed in detecting keypoints and extracting skeletal data, making it a strong candidate for real-time clinical use. Compared to other methods such as OpenPose [17,18], MediaPipe [19], DeepLabCut [20], AlphaPose [21], and PoseNet [22], YOLO Pose demonstrates superior performance in terms of computational speed and resource efficiency, making it more suitable for real-time applications on standard hardware. While OpenPose and AlphaPose achieve high accuracy, they generally require more powerful GPUs and longer processing times, which can limit their usability in clinical settings where immediate feedback is critical. MediaPipe offers fast processing but is often less accurate in complex poses, and DeepLabCut is primarily designed for animal pose estimation. PoseNet, although lightweight, tends to produce lower precision in joint localization compared to YOLO Pose.

YOLO Pose offers a lightweight and easily integrable solution, but this comes with a potential trade-off in precision.

This preliminary study investigates the suitability of YOLO Pose for detecting 17 keypoints relevant to physiotherapy assessments, balancing model complexity and real-time performance. Beyond evaluating the raw detection capabilities, we analyze several neural network variants and propose architectural modifications to enhance accuracy [23,24,25,26]. Moreover, we explore the design of a semantic framework to convert pose sensor data into clinically relevant insights, supporting more objective, data-driven evaluations of motor function.

To structure this investigation, we pose the following research questions:Is the YOLO Pose algorithm suitable for detecting physiotherapy-relevant keypoints from sensor data obtained with cameras?If it is not sufficient, can its precision be improved through neural network modifications?Is it possible to develop a semantic model that translates keypoint data into meaningful clinical parameters?

This work is structured into a theoretical framework, evaluation of YOLO, a comparative study of neural networks, and the development of a semantic for physiotherapy. Finally, conclusions and future research directions are presented.

## 2. Related Works

### 2.1. Evolution of Physiotherapy and the Role of Sensors

Historically, physiotherapy has focused on the treatment and rehabilitation of musculoskeletal and neuromuscular conditions, gradually integrating advanced technologies that optimize both patient assessment and intervention [27,28]. This development is particularly relevant in the use of movement analysis and sensor-based data capture technologies, which have radically transformed physiotherapy practice by providing new dimensions for the analysis and understanding of the patient’s body dynamics [4,5]. Historically, physiotherapy has concentrated on visual and manual assessments to determine the patient’s functionality, methods that are often subject to some degree of subjectivity. This subjectivity can lead to variability in interpretation and recommended treatments, especially in complex situations such as the assessment of brain damage [1]. In this neurological context, the limitations of manual evaluation are particularly evident, where small variations in movement can be indicative of significant neurological alterations [2]. The adoption of motion capture technologies, such as advanced video systems with optical sensors, has represented a revolutionary shift. These tools provide the ability to obtain precise and objective data about body movement, overcoming the challenges of evaluator variability and offering a means to quantitatively and reproducibly measure motor capabilities and neuromuscular dysfunctions [3]. These systems enable highly detailed and technical visualization of human movement kinematics and kinetics, which is crucial for understanding movement patterns and deviations from expected norms [6,7].

### 2.2. Clinical Assessment Scales in Physiotherapy

In the field of brain injury, precise movement assessment is indispensable. Quantitative movement data not only help to more reliably determine the degree of neurological impairment but also allow for the monitoring of patient progress over time, facilitating the modification and customization of treatments based on objective evidence [2]. These quantitative movement data are crucial for reliably determining the degree of neurological impairment and for tracking patient progress over time. This detailed follow-up is essential in neurological physiotherapy, where maximizing functional recovery and minimizing long-term sequelae are often primary objectives [3,14]. To establish a common framework, increase diagnostic accuracy, and develop evidence-based treatments [8], physiotherapists often use specialized assessment tools such as the Berg Balance Scale for balance, the Motor Index for strength and coordination, or the National Institute of Health Stroke Scale (NIHSS), which is particularly useful for assessing the severity of neurological damage in stroke patients. These tools allow for the specific quantification of motor and functional capabilities, providing less subjective parameters that are essential for determining the impact of brain damage and monitoring recovery [9]. In this context, the integration of specialized assessment scales, such as those mentioned above and others like the Functional Independence Measure (FIM) and the Functional Ambulation Category (FAC), becomes critical. These assessment tools are designed to specifically quantify motor and functional capabilities, providing objective parameters that help determine the impact of brain damage and monitor recovery progression.

It seems relevant to generically outline what each scale measures, as can be seen in Table 1.

### 2.3. Challenges in Mapping Sensor Data to Clinical Scales

This semantic and data-driven approach represents a significant advancement in physiotherapy, enabling more targeted treatments grounded in a deep understanding of the patient’s body dynamics. By adopting this systematic and evidence-based approach, physiotherapy can offer more precise assessments and personalized treatments, which are fundamental for the optimal recovery of patients with brain injury.

However, directly mapping raw motion capture data to these scales is not straightforward, as these scales often rely more on clinical observation and patient responses than on pure kinematic sensor data [3]. To bridge this gap, a systematic approach is needed to translate the quantitative movement measurements captured by technological tools into clinically meaningful evaluations [10].

In practice, the data obtained from motion capture must be interpreted through clinical knowledge to be effectively used in patient evaluations. This involves understanding the biomechanical implications of the data and how they relate to the specific clinical picture of the patient [2]. For example, variations in gait and posture captured through motion analysis may indicate progress in a patient recovering from a stroke. These variations need to be correlated with improvements or deteriorations observed in standardized assessments such as the FIM or FAC to validate the clinical relevance of the observed changes [11].

### 2.4. Applications of Computer Vision in Physiotherapy

The translation of joint keypoints and the timing in each position or movement into interpretable clinical information involves methods to quantify and assess movement quality within established clinical parameters. Although there are significant challenges in this translation due to the complexity of human movements and individual variability, some research has attempted to address these issues through the use of computer vision applied to data collected from optical sensors and deep learning models to improve the interpretation of motion data [4,13].

The complexity of this challenge lies in the variability and multidimensional nature of human movements, as well as the need to align these detailed kinematic data with clinical evaluation systems, which are often more subjective and based on direct observation. Some studies have explored the application of computer vision and deep learning models to improve the ability to accurately translate motion sensor data into clinical evaluations. For example, researchers such as [1] have developed methods that use deep neural networks to interpret motion sensor data and map them to assessable clinical outcomes, such as pain levels or movement ability. These models seek to find patterns within the motion data that reliably correlate with physical assessment indices used in physiotherapy.

Moreover, studies like that of [14] have investigated the use of motion capture technologies to provide quantitative metrics that can be integrated into electronic medical records and used to monitor rehabilitation progress more objectively and measurably.

On the other hand, [2] highlights how motion capture technology, combined with advanced analysis, can contribute to a better assessment of gait disorders in neurological patients. The ability to precisely quantify gait abnormalities and other motor functions is crucial for adjusting treatments and measuring their effectiveness over time.

These studies demonstrate the potential and challenges of integrating advanced measurement technologies into clinical practice, emphasizing the need to continue developing tools that can simplify and improve the accuracy of translating quantitative data into meaningful clinical assessments.

### 2.5. YOLO Pose for Movement Analysis

Computer vision, a fundamental discipline within artificial intelligence, can be a powerful tool to help interpret and understand the data collected from optical sensors about the movements performed by humans. This technology simulates human perception by capturing, analyzing, and processing images and has been widely implemented across various sectors such as medicine, security, and the automotive industry, significantly enhancing efficiency and accuracy in these fields [10]. In physiotherapy, computer vision represents an innovative and increasingly valued tool for diagnosing and treating patients, facilitating the personalization of therapies based on detailed and objective evaluations of human biomechanics [12].

Most existing pose estimation approaches represent the body as a set of undifferentiated keypoints without distinguishing specific anatomical segments. However, in physiotherapy and rehabilitation, the ability to separately analyze the motion of limbs and trunk is crucial for detecting asymmetries, compensatory patterns, or subtle deficits. Therefore, a segmentation approach that explicitly models each limb and the trunk could offer significant clinical advantages.

In physiotherapy practice, the implementation of cameras and specialized optical sensors allows for real-time capture of body movements. These devices not only increase the accuracy of movement measurements but also enable physiotherapists to make more accurate diagnoses and develop specific treatments tailored to the needs of each patient [4,5]. The ability to monitor patient movement effectiveness in detail aids in assessing treatment progress and verifying the recovery of motor skills after injuries or surgical interventions. Computer vision technology has proven especially useful in this context, offering solutions that can be easily integrated into daily clinical practice [14].

Specifically, YOLO, as mentioned earlier, is a system that allows real-time object detection and is known for its speed and efficiency, making it a valuable tool for the objectives of this work. The adaptation of this technology for pose estimation, known as YOLO Pose, has been particularly beneficial in areas that require accurate identification and tracking of body keypoints in image or video sequences. In physiotherapy, YOLO Pose is used to analyze human biomechanics in detail, which is crucial for gait assessment, injury rehabilitation, and the functional evaluation of patients [4].

YOLO Pose could allow physiotherapists to obtain detailed information about the position, orientation, and movement of different body parts, providing crucial data for the assessment of mobility and gait, where abnormal patterns may indicate neurological or musculoskeletal issues. This technology could also be highly relevant in injury rehabilitation, ensuring that recovery exercises are performed correctly to minimize the risk of relapses or new injuries. Additionally, functional assessment with YOLO Pose could help analyze how therapeutic interventions influence the patient’s ability to move in daily activities, thus facilitating the customization and adjustment of treatments [29,30].

The use of YOLO Pose in physiotherapy could not only improve the accuracy of treatment assessments compared to traditional methods but also provide a foundation for more targeted and evidence-based treatments. As this technology continues to develop, its application in physiotherapy is expected to expand, offering new opportunities to improve rehabilitation outcomes and optimize treatment plans. This advancement represents a significant shift in how physiotherapists approach assessment and treatment, leading to a substantial improvement in the quality of care and the effectiveness of therapeutic interventions [31].

This comprehensive approach to the application of YOLO Pose and computer vision in physiotherapy highlights the ongoing evolution of technologies and their potential to transform traditional practices, ensuring that patient care is both innovative and deeply grounded in detailed, scientifically validated analysis.

### 2.6. Ethical Considerations in the Use of Advanced Technologies

However, the use of advanced technology such as motion capture and artificial intelligence in physiotherapy raises important ethical considerations. Privacy and data confidentiality are crucial [32], as motion capture involves the collection of detailed information about the body and movements of patients, which is highly sensitive. According to the General Data Protection Regulation (GDPR) of the European Union, any data that can identify an individual must be protected with the utmost rigor. Institutions must implement robust security protocols to ensure that patient data are stored and transmitted securely, preventing unauthorized access and potential breaches.

Informed consent is another fundamental ethical principle [33]. Patients must be fully informed about how the technology will be used, what data will be collected, how they will be stored, and who will have access to them. This includes explaining the potential benefits of using motion capture technology in their treatment, as well as any possible risks. Patients must have the opportunity to ask questions and clearly understand this information before giving their consent.

The implementation of advanced technologies may exacerbate existing inequalities [34] in access to healthcare. Clinics and hospitals with fewer resources may not be able to acquire or maintain these technologies, leading to a gap in the quality of care provided to patients from different socioeconomic backgrounds. It is essential for developers and healthcare service providers to work toward making these technologies accessible and affordable for a wide range of institutions and patients, thus promoting equity in healthcare.

Physiotherapists and other healthcare professionals have the responsibility to use these technologies ethically and professionally, which includes being well-trained in their use and understanding their limitations. The integration of new technologies into clinical practice should be accompanied by continuous training programs to ensure that healthcare professionals can correctly interpret and apply the data generated by these technologies. Misinterpretation or incorrect use of these data could lead to erroneous diagnoses or inappropriate treatments.

The use of advanced technologies in physiotherapy must also respect patient autonomy, considering them active participants in their treatment and allowing them to have a voice in decisions that affect their care. Transparency in communication about how technologies will impact their treatment and possible outcomes is essential to foster trust and cooperation. Health institutions and regulatory bodies must continuously evaluate and monitor the use of motion capture technologies to ensure they adhere to ethical and professional standards. This includes assessing the clinical efficacy of these technologies and their impact on the quality of patient care. Periodic reviews and audits can help identify and correct ethical and operational issues [35,36].

### 2.7. Strategies for Improving Pose Estimation Algorithms

On the other hand, when analyzing the algorithm that allows the precise extraction of these parameters, it is important to consider the role of neural networks, especially convolutional neural networks (CNNs) [23,24]. These networks have proven to be extremely effective in computer vision tasks and have a structure that can be modified to improve their accuracy, efficiency, and overall capability. Strategies to improve neural networks, particularly YOLO Pose, include various techniques and methodologies based on current research.

Data augmentation is an important technique to improve neural networks. This process involves creating variations of existing training data, such as rotations, scale changes, and brightness adjustments, which helps the network learn more robust features and prevents overfitting [37]. Regularization is another relevant technique; methods like Dropout, L2 regularization, and batch normalization are used to improve the network’s generalization ability. For example, Dropout randomly deactivates a fraction of neurons during training to prevent overfitting [38,39].

The use of advanced optimizers such as Adam, RMSprop, and AdaGrad can accelerate training convergence and improve accuracy by adapting the learning rate according to the data characteristics and network architecture [40]. Experimenting with more complex or specialized network architecture can also lead to significant improvements. Residual networks and dense neural networks have shown improvements in accuracy by addressing the degradation problem and facilitating the flow of information through deep layers [25,41].

To specifically improve YOLO Pose, the resolution of input images can be increased, helping to capture finer details crucial for accurate pose estimation. Implementing specialized loss functions that more effectively penalize pose estimation errors can also improve performance [42,43]. Using pretrained models on large related datasets and fine-tuning these models with task-specific pose estimation data can significantly improve the accuracy and efficiency of YOLO Pose [44].

Updating the network’s backbone to more modern and efficient architectures like EfficientNet or MobileNetV3 can provide substantial improvements in accuracy and computational efficiency [45].

These strategies and improvements are supported by recent research demonstrating their effectiveness. For example, the work described in [25] on residual networks and [45] on EfficientNet has contributed to the development of more efficient and accurate network architectures. The use of advanced regularization techniques and adaptive optimizers has also shown significant improvements in neural network performance, according to studies by [38,40]. Transfer learning, discussed in [37], is another key area that has enabled pretrained models to rapidly improve on new specific tasks.

In summary, the following open issues identified in the review of the state of the art are addressed in this research: the lack of a standardized and objective method to map motion capture data to clinical assessment scales, which currently rely heavily on subjective observation; the variability inherent in traditional physiotherapy evaluations, leading to inconsistencies in diagnosis and treatment; the challenge of interpreting quantitative kinematic data in a clinically meaningful way without requiring advanced technical expertise; the limited integration of computer vision tools such as YOLO Pose into everyday clinical workflows due to technical and resource constraints; the need for real-time, cost-effective, and accurate pose estimation systems suitable for routine use in physiotherapy settings; and the ethical and regulatory concerns related to data privacy, informed consent, and equitable access to advanced technologies. This research proposes to address these challenges by evaluating the YOLO Pose algorithm as a lightweight and efficient solution and by developing a semantic framework that translates sensor-based motion data into clinically relevant indicators aligned with current assessment protocols.

## 3. Methodology

In order to address the key challenges identified in the literature—namely, the lack of standardized mappings between sensor-based kinematic data and clinical assessment scales (Section 2.3), the subjectivity and variability in traditional physiotherapy evaluations (Section 2.1 and Section 2.2), and the limited integration of lightweight, real-time pose estimation models into clinical practice (Section 2.4 and Section 2.5)—a series of experiments were designed, focusing on the evaluation and adaptation of the YOLO Pose model for use in the context of physiotherapy. This section outlines the overall approach, which includes both a preliminary visual assessment of the model and several structural modifications to the YOLOv8x-pose-p6 neural network.

The main objective is to analyze its ability to detect keypoints in images extracted from videos of rehabilitation exercises and assess its practical applicability. It is important to note that this section is dedicated exclusively to the methodology and approach followed, while the results obtained will be presented in the following section.

### 3.1. Evaluating YOLO Pose to Address the Identified Issues

A tailored visual evaluation was performed using YOLOv8x-pose-p6, the most advanced version of YOLO Pose, to verify the accurate identification of keypoints in a dataset of 1000 distinct images. This evaluation aimed to assess the model’s viability for physiotherapy applications, specifically for analyzing human movement in the context of rehabilitation exercises. By testing the model on this specific dataset, the study sought to determine its capacity to detect keypoints and identify posture variations, which are crucial for monitoring and improving rehabilitation progress. The visual assessment also allowed for the detection of potential errors that might not be visible through automated metrics, ensuring the model’s results met both technical standards and those for practical applications. Additionally, the evaluation addressed contextual and postural variations that could impact the model’s accuracy.

The selection of YOLO Pose for this study was made after performing a qualitative comparison against several state-of-the-art pose estimation frameworks, as summarized in Table 2. OpenPose [19] is a widely recognized open-source system that provides high anatomical precision through the use of part affinity fields (PAFs) to detect and associate body parts in multiperson scenes. However, its high computational demand and relatively low processing speed make it less suitable for real-time feedback in resource-constrained environments. MediaPipe [46] stands out for its ability to run efficiently on CPUs, including mobile devices, making it highly accessible. Yet, this lightweight approach compromises pose estimation accuracy in more complex postures. AlphaPose, based on the RMPE framework [21], delivers high accuracy in multiperson pose estimation by refining detection-then-estimation strategies, but also requires powerful GPUs and exhibits higher latency. In contrast, YOLO Pose [47] integrates pose estimation directly into the YOLO detection pipeline using object keypoint similarity loss, offering a favorable balance between accuracy and real-time performance. Its architecture enables single-pass, multiperson pose detection with significantly reduced inference time, making it particularly suitable for physiotherapy applications where fast and responsive interaction is essential.

In addition, previous experience within the PhyUM Research Center includes the use of OpenPose for automatic action recognition in karate kumite [48]. This experience has contributed to our understanding of the strengths and limitations of OpenPose, informing our selection of YOLO Pose for this specific clinical application.

The YOLOv8x-pose-p6 model [1] is one of the most advanced in pose estimation, characterized by high precision and the ability to detect multiple keypoints on the human body. Furthermore, the model’s attention mechanism improves its focus on relevant regions of the image, making it adaptable for precise pose detection in physiotherapy applications. The results from this preliminary study laid the groundwork for further optimization and modification of the model to better meet the needs of rehabilitation settings.

### 3.2. Creation of the Image Dataset

The image dataset used in this study was constructed with the intention of representing a wide variety of movement contexts and patient profiles. To that end, video frames were selected from professional athletes (e.g., gymnasts) performing controlled rehabilitation movements, as well as individuals with mobility impairments following physiotherapist-guided exercises. This selection aimed to introduce diversity in terms of physical condition, age, movement range, and execution quality. While the recordings were obtained from uncontrolled environments, including home settings and open spaces, this variability reflects the real-world heterogeneity found in physiotherapy practice—particularly in decentralized or home-based rehabilitation programs. Rather than constituting a limitation, such variability introduces valuable noise and contextual challenges, allowing the model to be stress-tested against common inconsistencies present in practical scenarios. Therefore, the dataset not only supports technical evaluation but also simulates realistic deployment conditions. Furthermore, the dataset was constructed to reflect a wide range of body types, mobility levels, and clothing conditions. Selected videos included individuals of various ages and body compositions—including children, older adults, athletes, and individuals with reduced mobility—wearing different types of clothing such as sportswear, casual attire, and loose garments. This heterogeneity was intentionally preserved to evaluate the model’s robustness under real-world variability and to minimize bias toward specific anatomical profiles. Although the dataset was not formally annotated by demographic or anthropometric category, its diversity supports the generalizability of the model to different physiotherapy populations.

Using several online videos, programs like [19,45] were employed to generate images from video frames.A simple program was created in Visual Basic to segregate the images from different videos and to organize the datasets into training, validation, and test sets. Approximately 1000 images were selected as the dataset.Once the necessary images for training were obtained, they were annotated using Computer Vision Annotation Tool (CVAT) software. CVAT is an open-source annotation tool designed for creating labeled images and video datasets, specifically for computer vision tasks (see Figure 1). CVAT allows users to draw bounding boxes, polygons, and keypoints on images, thus facilitating data preparation for machine learning models. More information on CVAT can be found in its official GitHub repository (https://github.com/cvat-ai/cvat (accessed on 14 February 2025)).

In the context of this research, CVAT was used to label bounding boxes for each body limb and trunk of individuals in the images. Additionally, keypoints within each limb were annotated. During the labeling process, the images were divided into six classes (five limbs and the trunk), rather than having a single “person” class. This strategy was employed to achieve higher precision in identifying and tracking each limb and the trunk. In the context of biomechanics or rehabilitation, deep learning models can benefit from data specificity. By having detailed labels for each body part, the models can learn to detect and analyze each component more effectively, potentially reducing ambiguity in complex poses. CVAT allows for generating labeled files in various formats but not in YOLO Pose format (see Figure 1). Therefore, a custom program was developed to convert the annotations from the CVAT-generated JSON file to the multiple .txt files and relative position conversions required by YOLO Pose.

**Figure 1 sensors-25-03436-f001:**
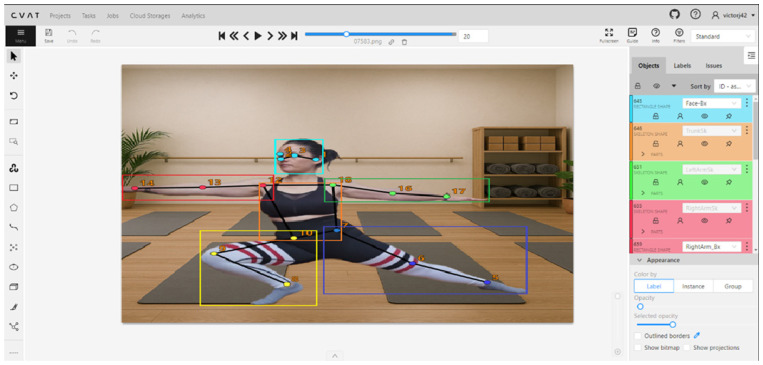
Example of an image labeling task using the CVAT program (released under the MIT License). The image was taken by the first author and features a female gymnast who signed an authorization to use her image in this paper. Different box colors represent different parts of the body.

Additionally, while different class labels were created, for the comparative study, a program was developed to group all data into a single bounding box (single “person” class) and a single line of coordinates for all keypoints, rather than six separate lines (one for each class).

This reduction in complexity was found to be more suitable for the study. Furthermore, the trunk keypoints were removed, as the shoulder and hip keypoints were used as their boundaries. Figure 2 illustrates an example of the visualized images resized to 640 × 640 pixels as required, with the final labeling in YOLO format. This process was programmed in Python using Google Colab.

Although the current dataset focuses primarily on body pose and movement patterns, future work will explore the inclusion of affective dimensions such as facial expressions. This is particularly relevant for assessing pain, motivation, or fatigue in physiotherapy contexts. Building upon our group’s prior work in facial emotion recognition in psychomotor settings [49], we plan to integrate multimodal datasets that combine body posture and facial cues to enhance semantic richness and support more holistic evaluation strategies.

### 3.3. Modifications to the YOLOv8-Pose-p6 Network

For this comparative study, a pretrained YOLO Pose network was not used. Instead, a custom image dataset was created in order to ensure full control over the data distribution, eliminate potential biases from pre-existing datasets, and tailor the training data to the specific domain and objectives of the study. By using a custom dataset, all network variants could be trained under identical conditions, enabling a fair and objective comparison of performance. The original YOLOv8-pose-p6 neural network was first employed to conduct an initial training session on this dataset. During this preliminary evaluation, the model exhibited difficulties in consistently detecting certain keypoints under physiotherapy-specific conditions, such as non-standard postures, partial occlusions, or subtle motion impairments—scenarios commonly encountered in rehabilitation settings. These limitations indicated that the default architecture, while effective in general-purpose applications, required further adaptation to meet the precision and robustness needed in clinical assessment tasks.

Subsequently, the network was modified in four different ways as identified in Table 3 to perform the same training with identical parameters, allowing for a rigorous evaluation of which variant produced the best performance. Conducting this comparative study using non-pretrained neural networks enables a more precise and controlled evaluation of the architectural modifications made to the network. By starting with a non-pretrained network and utilizing a customized dataset, biases and influences inherent in pretrained models based on external data are eliminated. This ensures that any observed improvement in performance is solely attributed to the introduced modifications and not external factors. Furthermore, training from scratch using a specific dataset provides more relevant and contextually applicable insights. Variations in performance between the different network modifications can be directly attributed to architectural changes, making it easier to identify the most effective configurations for the specific task of human pose estimation. This approach also allows for more precise adaptation of the network to the unique characteristics of the custom dataset, optimizing performance in real-world and context-specific scenarios, rather than relying on pretrained models that may have been trained in different contexts.

#### 3.3.1. Original Network Configuration (YOLOv8-Pose-p6)

The YOLOv8-pose-p6 network, designed for pose estimation, is characterized by:Number of Classes (nc): Configured for a single class, considering the person as the only object of interest.Keypoint Shape (kpt_shape): Determines 17 keypoints with coordinates (X, Y) and a visibility indicator.Model Scale: Depth, width, and maximum number of channels set to [1.00, 1.25, 512].Architecture:✓Backbone: Composed of convolutional layers and C2f blocks, with Spatial Pyramid Pooling—Fast (SPPF) to capture features at different scales.✓Head: Includes upsampling operations and concatenations to integrate information from different levels, culminating in a structure specialized in pose estimation.


#### 3.3.2. Modifications

In the YOLOv8-pose-p6 architecture, the backbone is composed of convolutional layers and C2f blocks, which progressively extract features from the input images. The head consists of detection layers that refine and output the estimated keypoints. Batch normalization (BatchNorm2d) layers stabilize learning by normalizing feature distributions, and activation functions such as ReLU and LeakyReLU introduce non-linearity to enhance the model’s capacity to capture complex patterns. Modifications were applied to different stages of the network to improve abstraction, spatial detail retention, and generalization (see Table 3).

### 3.4. Training Characteristics

The training was conducted on an NVIDIA A100 GPU in Google Colab, utilizing:Data Parallelism: 8 Dataloader Workers to optimize data loading.Automatic Mixed Precision (AMP): Reduces memory consumption without affecting accuracy.Data Augmentation: Real-time transformations applied with Albumentations to enhance model generalization.Batch Size: Adjusted based on available memory (batch size of 64).Real-Time Monitoring: Use of TensorBoard for process adjustment and optimization.

### 3.5. Training Hyperparameters

Five models were trained (one corresponding to the original YOLOv8-pose-p6 network and four incorporating different architectural modifications) using the following hyperparameters:Number of Epochs: 300, ensuring sufficient exposure to the data.Number of Runs: 5, to evaluate performance variance.Image Size: 640 × 640 px, balancing efficiency and small detail detection.Optimizer: AdamW with automatic learning rate and momentum adjustment.Data Augmentation: Application of Blur, MedianBlur, ToGray, and CLAHE to improve model robustness.Single-Class Training: Training with single_cls = True, optimizing detection for a single object type.

These modifications and configurations aim to enhance YOLOv8-pose-p6’s capability in pose estimation, optimizing learning and generalization in complex scenarios.

### 3.6. Data Extraction for a Common Semantics

To standardize the interpretation of movement data and ensure interoperability across different evaluation methodologies, a structured approach to data acquisition, processing, and transformation is implemented. This methodology allows for the integration of raw movement parameters into physiotherapy assessment scales through a hierarchical abstraction model.

#### 3.6.1. Data Acquisition: Controlled Recording Environment

A single-camera setup in a controlled room is used for patient monitoring. The environment ensures:Adequate space for unrestricted movement.Consistent lighting to avoid image distortions.Fixed high-resolution cameras to accurately capture movement.Specialized recording software for video analysis and data storage.

#### 3.6.2. Data Extraction and Key Parameters

During recording, the model extracts key movement parameters:Bounding Box Data: Position and size of the detected person.Keypoints: 17 skeletal keypoints (e.g., head, shoulders, elbows, wrists, hips, knees, ankles).Timestamps: Capturing motion in milliseconds.Future Considerations: Facial expression analysis and finer joint detection.

#### 3.6.3. Data Processing: Transformation into Physiotherapy Metrics

To facilitate clinical interpretation, the extracted parameters are transformed into physiotherapy evaluation metrics through a three-layer abstraction model, as summarized in Table 4.

This three-layer abstraction process enables bridging the gap between raw sensor data and clinically meaningful metrics. In the first layer, the system directly captures keypoints and trajectories. In the second layer, kinematic variables such as joint angles, speed, and body orientation are derived. In the final layer, these quantitative parameters are mapped onto established physiotherapy assessment scales, improving interpretability and clinical utility.

Each of the four neural network modifications (described in Section 3.6.3) was evaluated to determine its impact on the quality and reliability of data extracted for each abstraction layer. In particular:Modification 1 (Increased Depth) aimed to improve the extraction of complex spatial features at the first and second layers, enhancing the quality of keypoint localization and movement trajectory accuracy.Modification 2 (Subsampling Adjustment and Activation Functions) targeted the retention of fine-grained spatial details and optimized gradient flow, improving the computation of joint angles and velocities at the second layer.Modification 3 (Incorporation of Normalization) focused on increasing the stability of feature extraction across training, ensuring more consistent pose detection relevant to all three layers.Modification 4 (Normalization and ReLU in Final Stage) enhanced the reliability of outputs at the third layer, facilitating better mapping of extracted metrics to clinical scales.

Thus, each modification was assessed across the three-layer abstraction pipeline to analyze its contribution to the clinical interpretability of the generated data.

For each modified network (NC_PB, TS_FA, CN, CN_FA), all three abstraction layers were implemented and the corresponding clinical mappings (Motor Index, MAS, Berg, Barthel) were analyzed to determine the performance impact.

## 4. Results

This section presents the main findings derived from the application of the YOLO Pose neural network in the context of physiotherapy assessment, following the methodology described in this work and summarized in Figure 3. The reader will first find a preliminary evaluation of the base model, highlighting its overall performance and the limitations identified in accurately detecting human body keypoints.

Next, this section details the process of modifying and retraining the YOLOv8x-pose-p6 architecture. In total, five configurations were evaluated: the baseline model (original YOLOv8x-pose-p6 without modifications) and four modified versions corresponding to the architectural changes described previously (Increased Depth, Subsampling Adjustment and Activation Functions, Incorporation of Normalization, and Normalization and ReLU in Final Stage). Each configuration was trained and assessed using metrics such as accuracy and loss per epoch. A comparative analysis was conducted to assess the impact of each modification on pose estimation performance, supported by visualizations and a summary table of the most relevant parameters, as discussed in Section 4.3.2 and Section 4.3.3.

The section then introduces a system based on three levels of abstraction, which transforms the extracted data (e.g., position, angles, time) into clinically interpretable parameters. This transformation enables a connection between raw data and standardized physiotherapy evaluation scales, such as the Modified Ashworth Scale, the Barthel Index, and the NIH Stroke Scale (NIHSS).

Finally, the section provides practical examples of the system operating in real time, including strategies for classifying patient orientation and calculating motion angles. All of this contributes to an automated, accurate, and meaningful interpretation of physiotherapy-related movements, aimed at assisting healthcare professionals in clinical decision making.

### 4.1. Preliminary Evaluation of YOLO Pose to Address the Identified Issues

Upon applying this model to a set of images, it was observed that in certain cases and for specific postures, keypoints were not detected with sufficient accuracy (see Figure 4).

It is important to note that, in the majority of instances, the keypoints were correctly detected—approximately 83.5% of the images were considered to have all keypoints identified accurately. The complete results obtained can be utilized to provide reliable information regarding patient mobility, particularly in images where the individual is oriented frontally toward the camera.

Nevertheless, it was recognized that there was room for improvement in the model’s performance. Consequently, several architectural modifications were designed and evaluated through a comparative study to determine whether altering the original YOLOv8x-pose-p6 structure could enhance pose estimation accuracy and training stability. Table 5 summarizes the ten configurations tested, including the baseline model (C0), four primary independent modifications (C1 to C4), and several logical combinations of these modifications (C5 to C9).

Although multiple combinations were explored, the study primarily focused on evaluating the four independent modifications represented by configurations C1 to C4. This strategy was adopted to ensure a controlled and interpretable analysis of each individual change’s contribution to network performance. By isolating the effects of increased depth, spatial detail preservation, training stability, and final stage activation refinement, we aimed to accurately assess the impact of each architectural adjustment. Further combined configurations (C5 to C9) are considered as potential future extensions beyond the scope of the current systematic evaluation. The detailed results of the comparative analysis are presented in the following section.

### 4.2. Modification and Training of the YOLOv8x-Pose-p6 Neural Network

To evaluate the results obtained from the training of each network configuration, both the best model’s accuracy and loss values were considered, as well as the average metrics across all training epochs. Accuracy was chosen because it directly reflects the network’s ability to correctly predict the keypoints’ locations, a fundamental criterion in pose estimation tasks [13,14]. Loss values, specifically the total loss combining classification, localization, and keypoint regression errors, were also analyzed as a comprehensive indicator of training quality and convergence behavior [17]. As highlighted in the review of the state of the art, pose estimation studies typically prioritize keypoint detection accuracy and loss minimization as primary evaluation metrics, given their strong correlation with the model’s practical performance in real-world applications such as rehabilitation, biomechanical analysis, and clinical assessment [1,4,6]. Therefore, the selection of these metrics ensures consistency with existing research practices while allowing for a reliable comparative analysis of the proposed network modifications.

Figure 5 and Figure 6 show the evolution of accuracy and loss per epoch for the best model obtained, while Figure 7 and Figure 8 present this evolution for the average accuracy and loss.

### 4.3. Constructing a Common Semantics

To bridge the gap between raw pose estimation data and clinically interpretable physiotherapy metrics, a three-layer abstraction framework is proposed. This framework enables the progressive transformation of spatial and temporal data into parameters used by established assessment scales.

Table 6 provides a summary of the most relevant results obtained for comparison.

The semantic framework has been implemented through this three-layer architecture, starting from raw keypoints, progressing through calculated biomechanical parameters (e.g., range of motion, joint alignment), and culminating in clinical-scale mapping. Although the current study focuses on the structural design and technical feasibility of this framework, future work will involve validation with expert clinicians and real patient data to confirm the clinical interpretability and relevance of the semantic outputs. Figure 9 provides a visual overview of the intended outcome, illustrating how raw motion data are analyzed and translated into clinically meaningful insights in real time.

In the tables analyzed in this work, several common parameters were identified, such as strength, stability, mobility, pain, and coordination. In Table 7, a comparative analysis of some of those tables is shown.

With these common parameters in mind, strategies were studied to transition from a lower level of abstraction (the raw parameters obtained from the trained model) to the parameters used in physiotherapy scales and tables for evaluations. In this regard, three levels of abstraction were identified.

Additionally, to complement the analysis of loss and accuracy, Table 7 presents a summary of key performance metrics for each network configuration, including average precision, recall, and F1-score.

These metrics were computed on the validation dataset using the final model weights obtained at the end of training. The F1-score provides a balanced view of the model’s ability to accurately detect keypoints while minimizing false positives and false negatives. These results reinforce the selection of the best-performing model for integration into the semantic framework.

#### 4.3.1. First Abstraction Layer

To ensure the accuracy and reliability of data collection, it would be advisable to conduct all patient recordings in a controlled room. This environment guarantees consistent lighting conditions, fixed camera positioning, and a uniform background, which significantly reduce visual noise and external interferences. Such standardization is critical to improve the precision and reproducibility of pose estimation, allowing for meaningful comparisons across sessions and between subjects.

The parameters obtained through the defined model during the patient’s recording in the controlled room are:Bounding Box: Position (X, Y) and width and height of the bounding box for the “person” class.Keypoints: Position (X, Y) of 17 keypoints: 0: Nose, 1: Left Eye, 2: Right Eye, 3: Left Ear, 4: Right Ear, 5: Left Shoulder, 6: Right Shoulder, 7: Left Elbow, 8: Right Elbow, 9: Left Wrist, 10: Right Wrist, 11: Left Hip, 12: Right Hip, 13: Left Knee, 14: Right Knee, 15: Left Ankle, and 16: Right Ankle.Time: The recorded time in milliseconds when the above parameters were obtained.

#### 4.3.2. Second Abstraction Layer

Physiotherapy tables systematize crucial information such as range of motion and pain scores, facilitating the standardization of treatments and the comparison of results over time. It is imperative to use a unified language and specific terminology that enhances communication between specialists and patients, ensuring a clear interpretation of treatments and outcomes.

Scales such as the National Institutes of Health Stroke Scale (NIHSS), the Disability Rating Scale (DRS), the Barthel Index, the Lawton and Brody Scale, the Catherine Bergego Scale (CBS), the Functional Ambulation Classification (FAC), the Berg Balance Scale, the Shortened Berg Balance Scale, as well as the Modified Ashworth Scale (MAS) and the Motor Index (MI), systematize crucial information for diagnostics and follow-ups.

The adoption of conversions between quantitative and qualitative data, and vice versa, is vital for clinical interpretations. Common scales such as the Visual Analog Scale (VAS) for pain and the Karnofsky Scale for functional capacity help interpret results and monitor treatment progress. In physical evaluation, specific methods are applied to measure the patient’s strength, flexibility, and endurance, utilizing dynamic tables for continuous recording and tracking. These scales and tables allow for the standardization of treatments and the comparison of results, improving communication between specialists and facilitating a clear interpretation of treatments and outcomes (see Table 8).

However, the multiplicity of these tools can complicate data management and information integration, leading to the need for a common framework that can integrate and translate collected data into more essential and abstract parameters, such as those obtained through advanced technologies like computer vision.

The implementation of a unified system that translates data obtained through high-tech methods into these various scales and tables would not only simplify clinical information management but also maximize treatment effectiveness. This approach would allow physiotherapists to focus more on patient care and less on data administration, optimizing time and resources in clinical practice.

Based on the previous parameters and using basic geometric calculations, it is possible to determine rotation and inclination angles of the limbs.

At this level, ranges of positioning and angles for these limbs can be established to assess and analyze patient mobility.

Using the recorded time parameter for each position, it is possible to obtain the duration spent within the established angle ranges, as well as other physical parameters such as angular velocity, angular acceleration, movement trajectory, and variation of inclination over a given period.

#### The Importance of Setting the Orientation

Before performing calculations on positions and angles, it is necessary to determine whether the patient is facing forward, turned, or in profile relative to the evaluating camera. This preliminary classification is crucial for subsequent analyses, such as determining the rotation angles of the limbs, as it provides a reference for the general orientation of the body.

To achieve this, different models (as described below) were used to classify images of individuals into three categories: “Frontal”, “Turned”, or “Profile”. This classification is based on the position of body keypoints (shoulders and hips) and the bounding box of the person.

The data were divided into two sets: features (X) and labels (Y). The features included the positions of the keypoints and the bounding box, while the labels represented the orientation class (“Frontal”, “Turned”, “Profile”). The LabelEncoder function was used to transform categorical labels into numerical values, facilitating their use in classification algorithms.

The dataset was split into training and testing sets, reserving 20% of the data for testing. This split was performed randomly but reproducibly using a fixed random seed (random_state). The features were scaled using StandardScaler to standardize them (zero mean and unit variance). This improves the performance and convergence of certain machine learning algorithms.

A principal component analysis (PCA) was applied to reduce the feature set to three principal components, aiming to facilitate visualization and improve model efficiency. However, since this reduction led to significantly worse results compared to keeping all features, it was ultimately not applied. Figure 10 shows a graphical representation of dimensionality reduction to two features using PCA as an example.

A comparative study was conducted using various classification algorithms, including logistic regression, decision trees, random forests, naïve Bayes, k-nearest neighbors, and support vector machines. Details are provided next.

Each classifier has advantages and disadvantages in evaluating the dataset. Logistic regression is an efficient linear model for linearly separable data. Decision trees provide interpretability through structured rules in a tree-like model, while random forests combine multiple trees to improve accuracy and reduce overfitting. Naïve Bayes, based on Bayes’ theorem, is efficient with high-dimensional data and small training datasets. K-nearest neighbors (KNN) classifies based on the majority of its nearest neighbors but can be computationally expensive for large datasets. Finally, support vector machines (SVMs) find the optimal hyperplane to separate classes in high-dimensional spaces, making them particularly useful when classes are clearly separable.

Each classifier was trained using the obtained training set and evaluated with the test set. The accuracy of each classifier was calculated, and the results were printed. Classifiers that require scaled features, such as support vector machines, were trained and evaluated using the scaled features.

The accuracy of all classifiers was compared, and the best-performing model was selected. This model is considered the most suitable for classifying the orientation of individuals in images. The best-performing model is used to determine the patient’s orientation, which then serves as the basis for identifying the remaining keypoints.

Each classifier has specific characteristics that make them suitable for different types of data and problems, and the process of selecting the best model involves evaluating its performance on the dataset. Results obtained are compiled in Table 9.

Given the results, a decision tree model was selected for determining patient orientation, providing the foundation for subsequent abstraction layers. Based on the classified orientation and the detected keypoints, several biomechanical parameters were computed to quantify movement quality and patient stability. These parameters include:Inclination and Rotation Angles: Keypoints corresponding to the head, torso, arms, and legs are used to calculate inclination and rotation angles relative to anatomical axes. These angles (e.g., lateral head tilt, head rotation, lateral trunk tilt) allow for the evaluation of symmetry, posture control, and compensatory patterns (see Table 10, columns 1–3).Time: For each detected motion, the system computes the time spent within defined angular ranges (e.g., 0–30°, 30–60°, 60–90°). This provides an objective measure of endurance and postural stability, particularly important in balance and gait assessments.Angular Velocity and Acceleration: These parameters are derived by calculating the time derivatives of inclination and rotation angles. They quantify the speed and dynamic control of transitions between positions, which are critical for assessing neuromuscular coordination.Variation of Spatial Position Over Time: By tracking the changes in spatial position over successive frames, the system measures the stability of the patient’s posture and gait, identifying sways, tremors, or balance issues.With or Without Weight: Some assessments involve evaluating the patient’s movements under additional load (e.g., with weighted objects or resistance). This parameter indicates whether the performance differences observed are influenced by external force application, which is crucial for assessing strength and compensatory strategies.

#### General Solutions from Abstraction Layer 2

Based on the data extracted from Abstraction Layer 2, generic scales and formulas were proposed to determine the patient’s condition. Figure 11 illustrates a sample report automatically generated through the Biomechanical Abstraction Model (BAM) based on the data extracted at Layer 2.

In Figure 11, the patient’s orientation and the angles defined in the current abstraction layer are displayed in real time. The system marks the box in green if the patient was able to move through all the defined movement ranges for the parameters in Table 10, orange if the patient moved through more than one range, and red if they could only remain in one of the defined ranges.

With the data generated during the performance of the exercises, the score is continuously updated, which determines the patient’s condition. Once the exercises are considered complete, the system records the last updated score as the final evaluation.

#### 4.3.3. Third Layer

At this level, movement ranges can be established (these can be adjusted as needed and serve as a regulatory parameter for patient evaluation). Table 10 presents an example of these considerations, bringing us closer to a translator for the tables used by physiotherapists to assess patients.

The third level is based on transforming, analyzing, and converting the information from the previous layer into specific tables and scales for patient evaluation, according to specific characteristics and the degree of specialization.

To map the extracted biomechanical parameters to standardized clinical evaluations, several established physiotherapy and neurological assessment scales were used. Table 11 summarizes the clinical purpose of each scale, the analyzed information extracted from pose estimation data, and the method used for scoring and interpretation.

## 5. Discussion

This section discusses the main findings of the study, analyzing the effects of the architectural modifications applied to the YOLOv8-pose-p6 model and their impact on pose estimation performance. The previous sections described the evolution of physiotherapy practices with technological integration, reviewed existing pose estimation algorithms, detailed the development of the Semantic Pose Analysis Pipeline (SPAP), and presented experimental results from different network configurations. Building upon this foundation, the following discussion critically examines the improvements achieved, evaluates the practical implications for clinical physiotherapy environments, and identifies potential directions for future enhancements.

### 5.1. Architectural Improvements in Pose Estimation

The experimental results from the various configurations of the YOLOv8-pose-p6 network reveal the strong influence of architectural modifications on pose estimation performance. Notably, in NETWORK 2, deepening the backbone by increasing the number of repetitions in convolution and C2f blocks led to a substantial improvement in accuracy and loss reduction. A deeper backbone enables the model to learn more abstract and nuanced features, crucial for recognizing subtle postural differences in fields like sports or physical rehabilitation.

Similarly, increasing the number of channels in the head layers also improved accuracy. This suggests that greater information density enhances the model’s ability to process and synthesize complex inputs, improving the precision of posture predictions.

### 5.2. Superior Performance of NETWORK 3

Among the tested models, NETWORK 3 outperformed the others. This is attributed to an effective combination of less aggressive subsampling and optimized activation functions (ReLU and LeakyReLU). A lower subsampling rate preserves spatial information across layers, which is especially valuable for detailed pose recognition.

Moreover, ReLU and LeakyReLU improved training efficiency. ReLU accelerates learning with its simple non-linearity, while LeakyReLU mitigates vanishing gradients by allowing slight gradient flow when units are inactive. This hybrid activation strategy enhances gradient propagation and model learning capacity.

### 5.3. Design of the Evaluation Environment in Physiotherapy

Implementing a pose estimation model in clinical settings requires a well-designed physical and technological environment. The space must allow for unrestricted movement, proper lighting, and optimal camera placement for multiangle coverage. High-resolution cameras are essential to ensure detailed posture tracking.

Complementing this setup, appropriate tools and exercise equipment must be available to replicate everyday movements and support physiotherapy protocols. All elements contribute to a precise and replicable evaluation process.

### 5.4. Technological Infrastructure and Parameter Analysis

The environment must be equipped with an advanced system for video analysis and data recording. This includes storage infrastructure capable of handling large video datasets and analytical software that extracts relevant movement parameters such as joint angles, timing, and facial expressions indicating pain.

Emerging capabilities, such as detecting finger and toe movements or analyzing hand joint articulation, could further enrich physiotherapy assessments. These future enhancements point toward increasingly granular and useful clinical insights.

### 5.5. Integration with Clinical Evaluation Tools

The interpretation of data should align with existing physiotherapy scales and evaluation tables. These tools help standardize assessments and enable longitudinal comparisons. For example, mobility and balance data derived from angles and positions can be mapped onto clinical scores for strength, pain, and coordination.

The explicit separation of limbs and trunk during labeling contributed to enhancing the clinical interpretability of the results. This approach allowed the pose estimation system to generate data that aligned more directly with the dimensions assessed by traditional clinical tools, such as strength evaluation per limb (Motor Index) or balance control (Berg Balance Scale). The anatomical specificity obtained during the labeling phase proved to be critical for bridging raw pose data with clinically meaningful abstraction levels.

### 5.6. Levels of Abstraction and Clinical Usability

A critical aspect of this work is the translation of raw pose estimation data into clinically actionable parameters. By leveraging a multilayer semantic abstraction framework, low-level data such as joint positions and body angles are transformed into higher-level constructs like mobility ranges, balance indicators, or postural stability metrics. This hierarchical organization enhances interpretability and facilitates communication with clinicians, improving the practical utility of pose estimation in physiotherapy assessment.

Although clinical validation with real patient data remains part of future work, we have taken a first step by presenting the semantic framework to physiotherapy professionals at Grupo 5 Centro Integral de Atención Neurorehabilitadora (CIAN) in May 2024 during an online meeting of the ReBrain project led by Kapres Technology. Their feedback was instrumental in refining the definitions and thresholds used in the framework, helping ensure its alignment with real-world rehabilitation practices.

In addition, future work will incorporate facial expression analysis to enable a more comprehensive semantic model that can integrate affective and cognitive indicators alongside motor assessments. This would expand the framework’s applicability to a broader range of physiotherapy evaluation scales.

We also envision training a neural network that uses the parameters extracted at the second semantic layer (e.g., range of motion, asymmetries, joint velocities) as input, with the outputs corresponding to clinical scores assigned by expert therapists using validated scales (e.g., Berg Balance Scale, Barthel Index). Such a model would facilitate automatic mapping from biomechanical features to clinical interpretations, thus closing the loop between raw sensor input and informed clinical decision making.

A summary of the current system components and the planned development trajectory is presented in Figure 12, which illustrates the semantic processing workflow and the roadmap toward clinical integration.

Furthermore, the ethical design of the system will be guided by the CARAIX framework [50], which promotes human-centered AI in psychomotor technologies. This includes ensuring transparency, patient autonomy, and contextual sensitivity in future deployments. Privacy-preserving strategies—such as real-time anonymization, encrypted data handling, and compliance with GDPR and institutional ethics protocols—will be integrated into the system’s architecture. These measures are essential to support clinical acceptance and to ensure that ethical and legal considerations are addressed from the design phase.

### 5.7. Scalability and Deployment Scenarios

The proposed system is designed with scalability and adaptability in mind, supporting deployment across a broad spectrum of physiotherapy scenarios, from large clinical facilities to decentralized home-based environments. In institutional settings—such as rehabilitation hospitals, outpatient clinics, or physiotherapy centers—the system can be deployed on local servers or integrated into existing healthcare IT infrastructures. Using centralized processing units or secure cloud platforms, it is possible to concurrently monitor multiple patients, store anonymized motion data, and allow for remote clinician access and review. This enables efficient resource allocation and supports tele-rehabilitation workflows, particularly beneficial in post-acute care or high-demand contexts.

For home-based physiotherapy, where infrastructure and supervision are typically limited, the system offers a lightweight and accessible solution. The model architecture has been optimized for real-time inference (~25 ms per frame on an RTX 3060) and can operate on consumer-grade laptops, tablets, or embedded edge devices equipped with standard webcams. This enables autonomous evaluation by the patient in domestic settings, promoting continuity of care and patient engagement while minimizing the need for in-person visits. The increasing availability of affordable hardware (e.g., NVIDIA Jetson Nano, Raspberry Pi with Coral TPU, or mid-range laptops) further enhances feasibility.

Moreover, the system’s modular structure allows selective activation of components depending on the computational capacity available (e.g., disabling real-time NLP or visual feedback modules in low-power deployments). This adaptive behavior ensures that both performance and usability can be tailored to the operational context. In future iterations, support for mobile deployments and integration with secure patient portals or electronic health record (EHR) systems may further enhance scalability, facilitating longitudinal monitoring and data-driven personalization of therapy protocols.

### 5.8. Common Pose Estimation Errors and Mitigation Strategies

Pose estimation models are known to suffer from common errors such as inaccurate keypoint placement, confusion between left and right limbs, and sensitivity to occlusion or extreme poses. In our study, these issues were observed particularly in non-frontal views and in images with low contrast or atypical clothing. To address these challenges, we applied several mitigation strategies:Incorporating diverse training data with varied lighting, orientations, and morphologies.Modifying the network architecture to preserve spatial information and reduce feature loss through downsampling.Applying anatomical plausibility checks in post-processing to identify inconsistent joint configurations.

These steps collectively improved robustness, but future work will explore the integration of temporal smoothing across video sequences and dynamic confidence-based filtering.

### 5.9. User Interaction and Workflow Integration

The proposed system is intended to serve as a clinical decision-support tool, aiding physiotherapists in evaluating patient movement using pose estimation and semantic abstraction. In a typical usage scenario, the patient performs predefined rehabilitation exercises in a controlled environment (e.g., a physiotherapy clinic or lab), guided by the clinician. A single RGB camera captures the session, and the system processes this input in real time, extracting key biomechanical parameters.

The physiotherapist interacts with the system through a graphical interface that displays clinically relevant values such as joint angles, posture stability, and motion completion, using intuitive visual cues (e.g., color-coded indicators and progress bars). This allows for efficient assessment and documentation, reducing subjectivity and supporting longitudinal tracking.

Patients do not interact directly with the system, ensuring ease of use and accessibility regardless of digital literacy. In future versions, the interface will be further refined using human-centered design principles and validated through usability studies with practicing clinicians. Voice-guided interaction via NLP and simplified feedback modules are also under consideration to support home-based rehabilitation.

Ultimately, the value of this system lies in its potential to enhance clinical decision making by transforming qualitative, observational assessments into quantifiable, reproducible metrics. Traditional physiotherapy evaluations often rely on subjective judgment, which can vary between practitioners and sessions. By providing objective indicators—such as joint angles, asymmetry detection, movement range classification, and angular velocity—the system enables more precise tracking of patient progress, earlier identification of functional limitations, and better-informed adjustments to therapy protocols. These outputs can be mapped to standard clinical scales, supporting longitudinal comparison and interdisciplinary communication. For example, a patient’s inability to reach specific motion thresholds can be flagged automatically and linked to predefined action plans. While formal outcome-based validation will require clinical studies, this work establishes the groundwork for integrating real-time biomechanical analysis into physiotherapy workflows, ultimately supporting more personalized, data-driven care.

## 6. Conclusions

The study demonstrates that strategic architectural enhancements to YOLOv8-pose-p6—especially the combination used in NETWORK 3—can significantly improve pose estimation accuracy. The refined architecture, optimized activation functions, and improved spatial information retention collectively contribute to better performance.

Moreover, the integration of pose estimation models into physiotherapy demands a thoughtful design of the evaluation environment, robust technical infrastructure, and alignment with clinical tools and practices. A hierarchical abstraction of raw data is essential for clinical interpretation and effective decision making. In this paper we have proposed a 3-layer approach called the Semantic Physiotherapy Abstraction Framework (SPAF), consisting in:Layer 1: Extraction of raw pose data (keypoints and bounding boxes) from camera input using YOLO Pose.Layer 2: Computation of biomechanical parameters (angles, velocity, orientation) through geometric analysis and machine learning.Layer 3: Mapping of computed parameters to standardized clinical scales (e.g., MAS, MI, BBS), enabling objective physiotherapy assessment.

Future research should explore more refined models, improved clinical validation, and the incorporation of additional parameters (e.g., finger and foot articulation) to support deeper and more individualized patient assessments. Additionally, affective components such as facial expression analysis and emotion recognition should be integrated into the framework to assess discomfort, pain, or emotional responses, enabling a more empathetic and holistic physiotherapy evaluation.

## 7. Limitations and Future Work

This study represents a foundational step toward integrating computer vision technologies into physiotherapy assessment through the use of pose estimation and a structured semantic abstraction framework. The results demonstrate promising performance in detecting keypoints and mapping movement data to clinically relevant metrics. However, several limitations and opportunities for improvement remain. As an initial exploratory step, the semantic framework was presented to physiotherapy professionals at Grupo 5 CIAN, whose feedback helped refine the abstraction levels and clinical relevance of the proposed mappings.

The current system has not yet been validated in real clinical environments with actual patients; instead, it was tested on a curated dataset compiled from publicly available videos, selected to reflect a range of rehabilitation scenarios under diverse, though uncontrolled, conditions. Additionally, the model currently relies solely on RGB optical input, which may be sensitive to lighting variation, occlusion, or patient clothing. Future iterations may benefit from incorporating complementary data sources such as depth sensors or inertial measurement units (IMUs) to enhance robustness and accuracy.

High-level semantic components—including facial expression analysis to infer pain or emotional state, and natural language processing (NLP) to support voice-guided interaction or interpret clinician input—are conceptually proposed but not yet implemented. These modules have the potential to significantly enhance user engagement, accessibility, and clinical relevance, particularly in home-based or unsupervised rehabilitation contexts. Similarly, formal usability testing with physiotherapists has not yet been performed, representing a key step for future validation and refinement.

To address these limitations, future work will focus on the development of a transition model between the second and third abstraction layers, mapping biomechanical data (e.g., limb angles, positions, and motion characteristics) into precise clinical indicators using advanced data analysis and machine learning techniques. This model will be informed by clinical expert feedback and evaluated using real assessments from professionals, serving as a bridge toward validation in real-world rehabilitation scenarios. The semantic framework will also be extended to cover additional physiotherapy tools beyond those already mapped (e.g., Motor Index, MAS, BBS, NIHSS, FAC, CBS, Lawton and Brody, and DRS), enabling broader and more standardized patient evaluation.

Furthermore, transfer learning strategies will be explored to facilitate adaptation to different patient profiles and rehabilitation contexts. Optimization for real-time processing will be prioritized to ensure responsiveness in live settings, and integration with other medical devices and electronic health systems will be considered to provide a more comprehensive and clinically useful assessment. Finally, a user-friendly software platform will be developed based on human-centered design principles, with the goal of supporting clinical adoption and maximizing practical usability through collaborative evaluation with healthcare professionals.

## Figures and Tables

**Figure 2 sensors-25-03436-f002:**
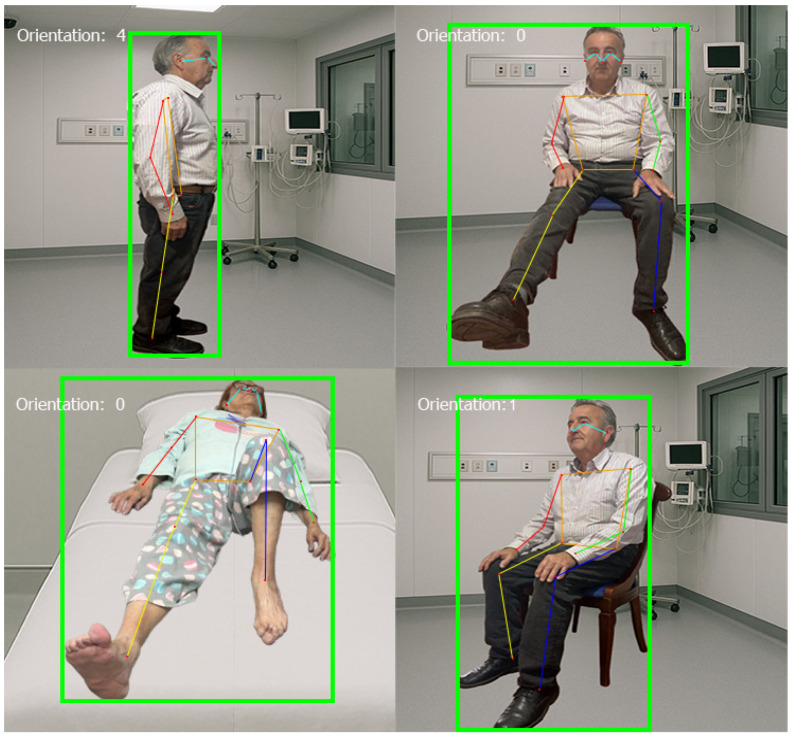
Example of images resized to 640 × 640 pixels with properly scaled keypoints and bounding boxes in YOLO format. The image was captured by the first author and depicts two retired individuals performing recommended physiotherapy exercises. Both participants provided written consent for the use of their image in this paper.

**Figure 3 sensors-25-03436-f003:**
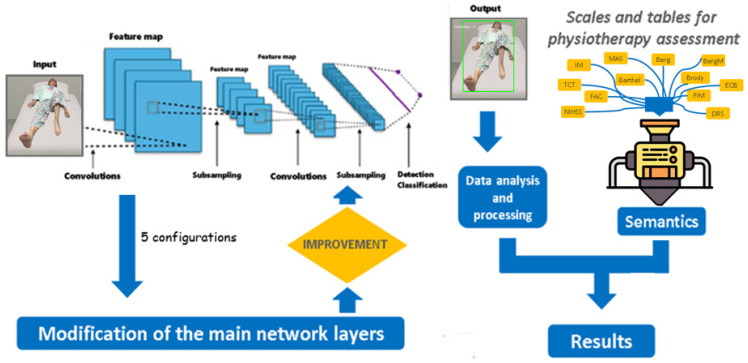
Summary of the methodology used to fine-tune the YOLO Pose network and obtain a common semantic framework for evaluating individuals with reduced mobility. The woman in the picture has signed a consent form authorizing the use of her image for the purposes of this study.

**Figure 4 sensors-25-03436-f004:**
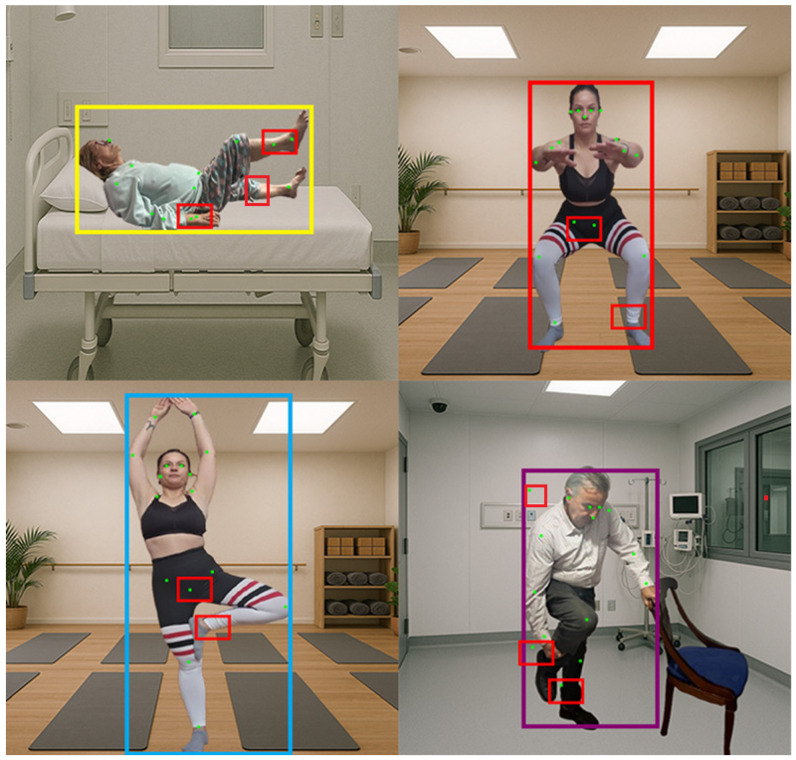
Use of the YOLOv8x-pose-p6 model in some images where keypoints were not detected properly. The images, captured by the first author, depict individuals engaged in gymnastics or physiotherapy exercises. All participants provided written informed consent for the use of their images in this research.

**Figure 5 sensors-25-03436-f005:**
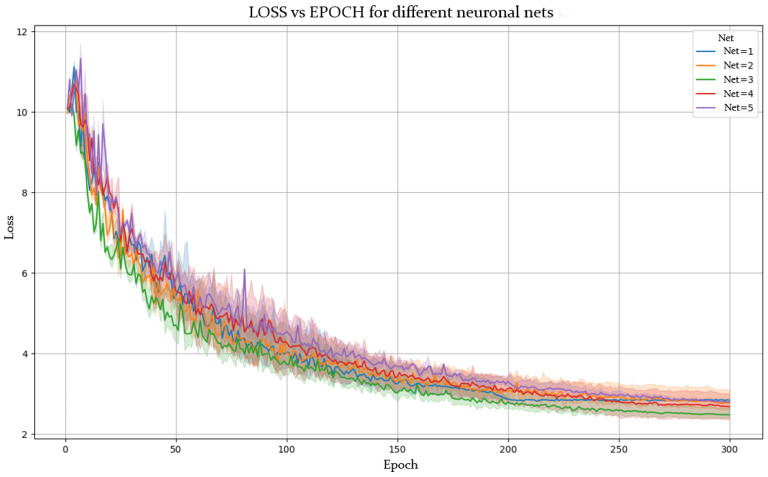
Evolution of the loss of the best model obtained for networks C0 to C4 specified in Table 5.

**Figure 6 sensors-25-03436-f006:**
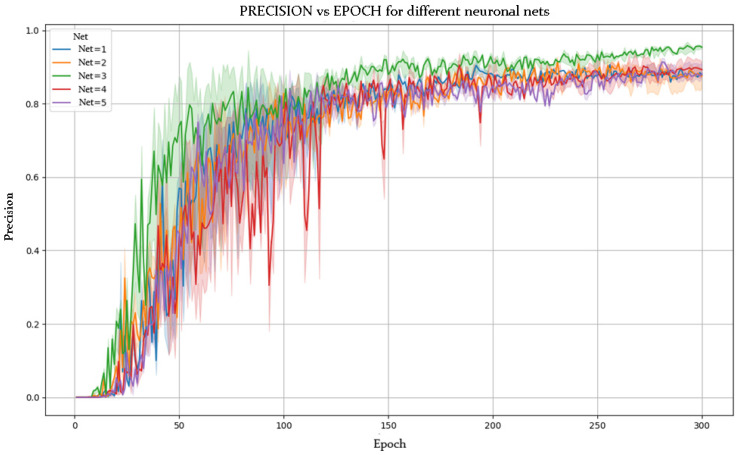
Evolution of the accuracy of the best model obtained for networks C0 to C4 specified in Table 5, according to the training epoch.

**Figure 7 sensors-25-03436-f007:**
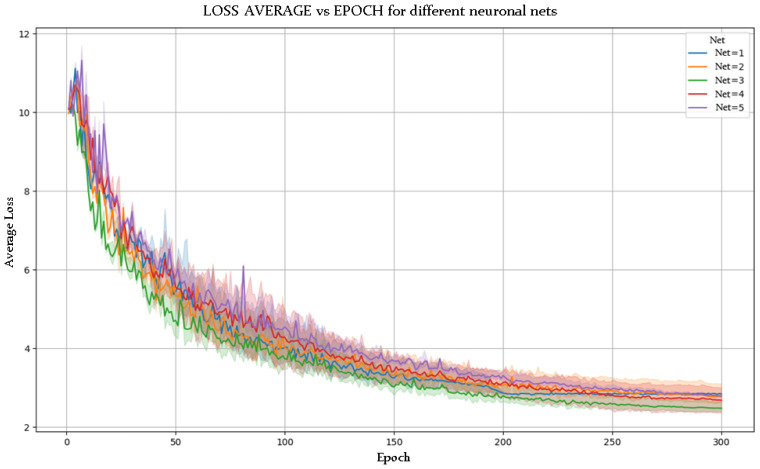
Evolution of the average loss obtained for networks C0 to C4 specified in Table 5, according to the training epoch.

**Figure 8 sensors-25-03436-f008:**
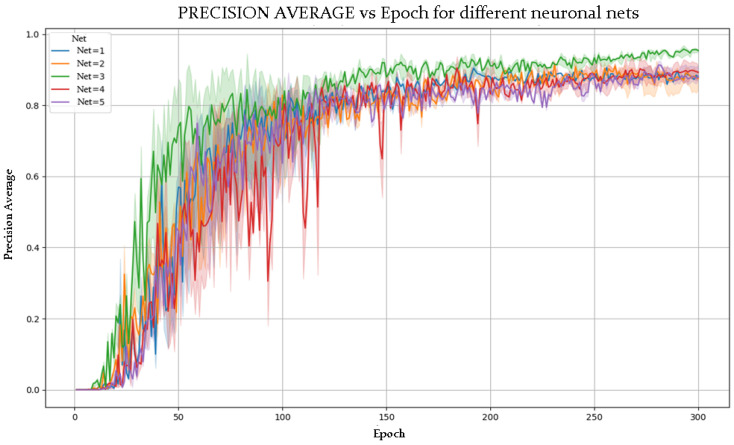
Evolution of the average accuracy obtained for networks C0 to C4 specified in Table 5, according to the training epoch.

**Figure 9 sensors-25-03436-f009:**
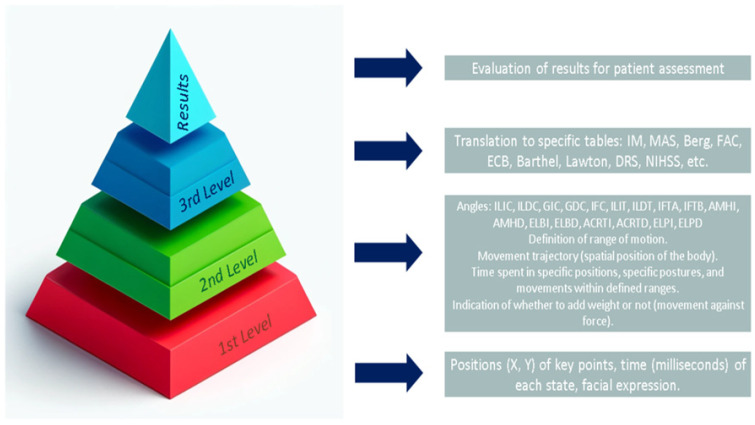
Visual representation of the evolution from the most abstractlevel (Layer 1) to the most concrete level (Layer 3), where a score is obtained in three different physiotherapy scales.

**Figure 10 sensors-25-03436-f010:**
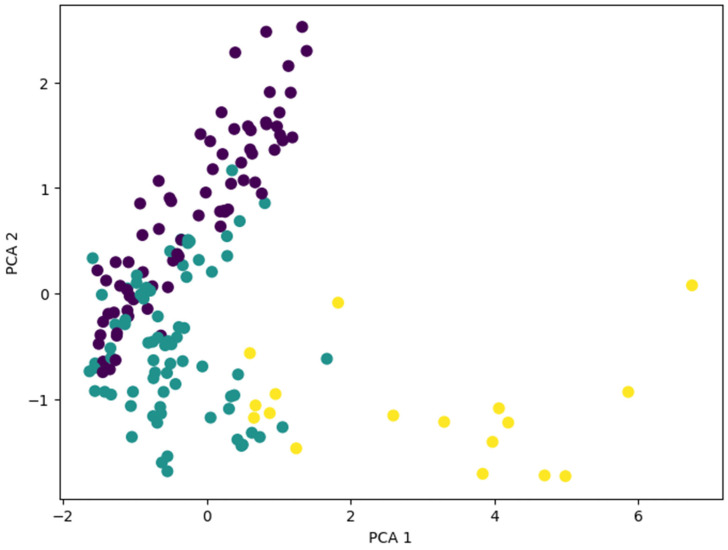
Dimensionality reduction to two dimensions of the four main features to determine the patient’s orientation.

**Figure 11 sensors-25-03436-f011:**
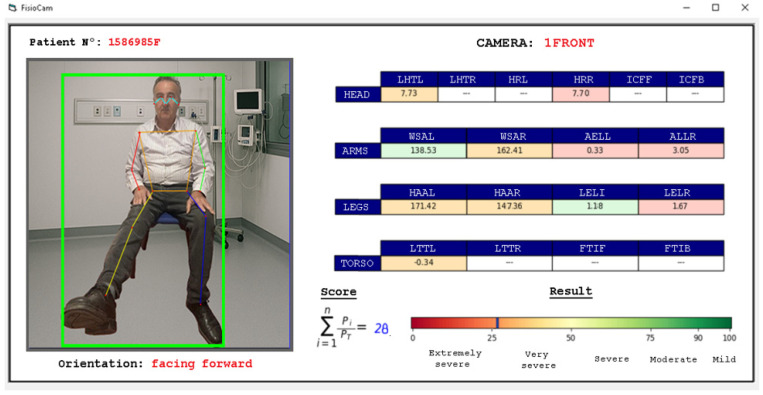
Software window displaying how it is automatically generated in real time based on the data extracted in Abstraction Layer 2.

**Figure 12 sensors-25-03436-f012:**
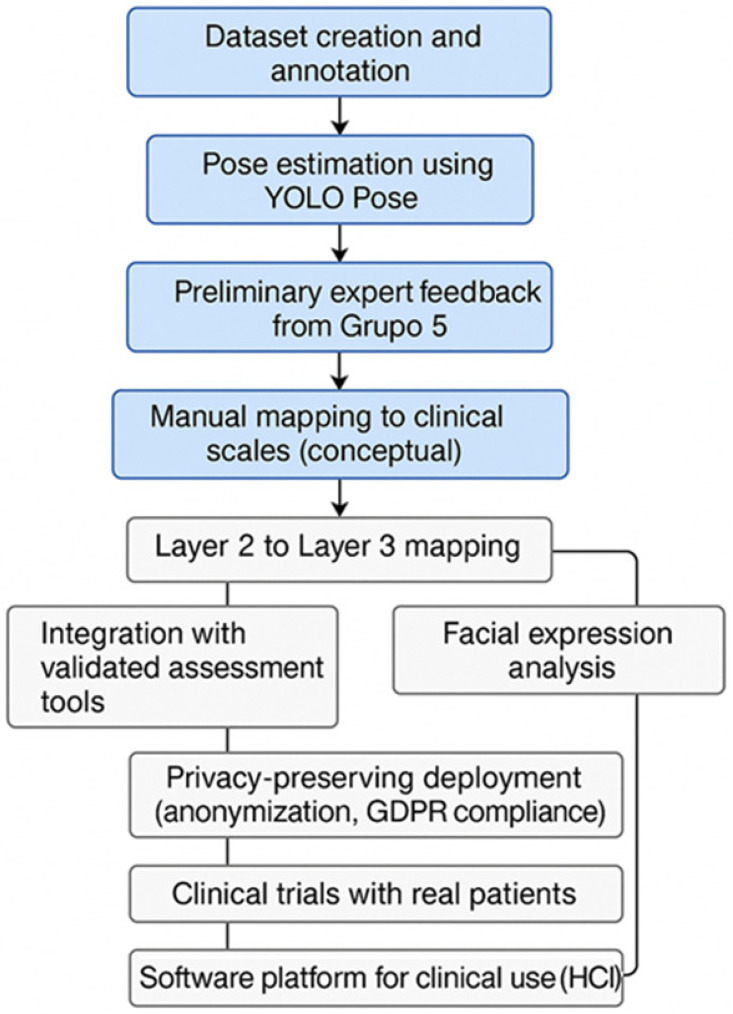
Semantic processing workflow and development roadmap of the proposed system. Modules in blue represent components that have already been implemented, while modules in gray correspond to planned future developments. The diagram reflects the hierarchical structure of the semantic model and its intended clinical integration, including affective computing, human-centered design, and ethical deployment strategies.

**Table 1 sensors-25-03436-t001:** Comparison of Clinical Assessment Scales Relevant to Mobility Evaluation in Physiotherapy.

Scale	Assessed Dimension	Measurement Type	Objectivity Level	Primary Functional Domain	Possible Correlation with Keypoints
Berg Balance Scale	Static and dynamic balance	Observational (scoring)	Medium	Postural control and stability	High: posture, center of gravity, sway
Motor Index	Muscle strength and motor control	Observational (ordinal scale)	Medium–high	Muscle strength and coordination	Moderate: range of motion, segment alignment
NIHSS	General neurological deficit (post-stroke)	Observational with specific tasks	Medium	Severity of neurological damage	Variable: depends on the item, some may map to fine or gross movements
FIM	Independence in daily living activities	Multidimensional functional scale	Medium	General functionality and independence	Low–Moderate: broad movements in ADLs, full functional patterns

**Table 2 sensors-25-03436-t002:** Comparative overview of state-of-the-art 2D pose estimation models relevant to physiotherapy applications.

Model	Accuracy	Speed (FPS)	Device Requirements	2D/3D	Open Source	Suitability for Rehab
OpenPose	High	Low	GPU required	2D	Yes	High (precise joints)
MediaPipe	Medium	High	Works on CPU	2D	Yes	Medium (lightweight)
AlphaPose	High	Medium	GPU recommended	2D	Yes	High (multiperson)
YOLO Pose	Medium–high	Very high	GPU/Edge support	2D	Yes	High (real-time)

**Table 3 sensors-25-03436-t003:** Summary of Architectural Modifications Applied to YOLOv8-pose-p6 for Optimizing Pose Estimation Performance.

Modifications	Network Variant	Changes Introduced	Objective
**1**	YOLOv8-pose-p6_NC_PB	Increased repetitions in C2f blocks (3→4 and 6→8); Increased channels in last head layer (1024→1280)	Enhance abstraction capacity and deep feature fusion
**2**	YOLOv8-pose-p6_TS_FA	Reduced subsampling in the backbone; Replaced activation functions with ReLU and LeakyReLU	Improve spatial detail retention and training efficiency
**3**	YOLOv8-pose-p6_CN	Integrated BatchNorm2d layers in the network head	Improve training stability and enable higher learning rates
**4**	YOLOv8-pose-p6_CN_FA	Added BatchNorm and ReLU in the final stage	Improve final stage activation and overall generalization

**Table 4 sensors-25-03436-t004:** Three-Layer Abstraction Model for Transformation of Pose Data into Physiotherapy Metrics.

Abstraction Layer	Description	Examples
**First Layer: Direct Data Extraction**	Extraction of raw pose information from keypoints and bounding boxes.	Identification of anatomical keypoints; recording of movement trajectories.
**Second Layer: Computational Analysis**	Processing raw data to compute derived biomechanical parameters.	Calculation of joint angles, range of motion, velocity, acceleration; orientation classification via ML models (decision trees, SVM, random forest).
**Third Layer: Mapping to Clinical Scales**	Transformation of computed parameters into physiotherapy evaluation frameworks to facilitate standardized clinical assessment.	Motor Index (IM), Modified Ashworth Scale (MAS), Berg Balance Scale, Barthel Index.

**Table 5 sensors-25-03436-t005:** Summary of Network Configurations and Applied Architectural Modifications for Pose Estimation Improvement.

Code	Description	Applied Modifications	Purpose
**C0**	Baseline YOLOv8x-pose-p6	None	Serve as performance reference.
**C1**	Increased Depth (NC_PB)	Increased C2f repetitions and head channels	Enhance feature abstraction and deep fusion.
**C2**	Subsampling Adjustment + Activation Functions (TS_FA)	Reduced subsampling; ReLU and LeakyReLU activations	Improve spatial detail retention and training efficiency.
**C3**	Head Normalization (CN)	Added BatchNorm2d layers in the head	Stabilize training and improve generalization.
**C4**	Final Stage Normalization + ReLU (CN_FA)	Added BatchNorm and ReLU at final stage	Improve final activation quality and output stability.
**C5**	Increased Depth + Subsampling Adjustment	Combined C2f depth increase and subsampling reduction	Maximize feature richness and spatial accuracy.
**C6**	Increased Depth + Head Normalization	Combined depth increase with head normalization	Improve deep feature learning and training stability.
**C7**	Subsampling Adjustment + Head Normalization	Combined subsampling reduction and head normalization	Enhance spatial extraction while stabilizing the network.
**C8**	Subsampling Adjustment + Final Stage Normalization	Combined subsampling reduction with final normalization	Preserve fine details and reinforce output stability.
**C9**	Increased Depth + Subsampling Adjustment + Head Normalization	Combined depth increase, subsampling adjustment, and normalization	Maximize abstraction, spatial retention, and robustness.

**Table 6 sensors-25-03436-t006:** Provides a summary of the most relevant parameters obtained for comparison.

NET	LOSS				PRECISION			
	Min	Average	Max	σ	Min	Average	Max	σ
**1**	2.812	4.485	1.128	1.872	0.000	0.646	0.917	0.308
**2**	2.595	4.005	1.081	1.597	0.000	0.708	0.938	0.262
**3**	2.390	3.743	1.069	1.632	0.000	0.779	0.969	0.269
**4**	2.356	4.162	1.070	1.829	0.000	0.675	0.954	0.292
**5**	2.778	4.334	1.169	1.801	0.000	0.690	0.946	0.275

**Table 7 sensors-25-03436-t007:** Per-keypoint precision, recall, and F1-score obtained from the best-performing YOLO Pose model evaluated on the validation dataset.

Class/Keypoint	Precision	Recall	F1-Score
Head	0.91	0.9	0.9
Neck	0.85	0.89	0.87
Right Shoulder	0.87	0.86	0.86
Left Shoulder	0.87	0.89	0.88
Right Elbow	0.92	0.91	0.91
Left Elbow	0.91	0.84	0.87
Right Wrist	0.93	0.91	0.92
Left Wrist	0.86	0.9	0.88
Right Hip	0.89	0.87	0.88
Left Hip	0.85	0.85	0.85
Right Knee	0.87	0.93	0.9
Left Knee	0.9	0.87	0.88
Right Ankle	0.85	0.85	0.85
Left Ankle	0.87	0.85	0.86

Estimated latency: ~24 ms per frame (measured on RTX 3060 GPU).

**Table 8 sensors-25-03436-t008:** Comparative Analysis of Physiotherapy Assessment Scales Based on Common Clinical Parameters. **✔** Parameter is explicitly assessed by the scale. **❌** Parameter is not assessed by the scale.

Assessment Scale	Strength	Stability	Mobility	Pain	Coordination
**NIH Stroke Scale (NIHSS)**	**✔**	**✔**	**✔**	**✔**	**✔**
**Barthel Index**	**✔**	**✔**	**✔**	**❌**	**❌**
**Modified Ashworth Scale (MAS)**	**✔**	**❌**	**❌**	**❌**	**❌**
**Berg Balance Scale (BBS)**	**❌**	**✔**	**✔**	**❌**	**❌**
**Motor Index (MI)**	**✔**	**❌**	**❌**	**❌**	**❌**
**Functional Ambulation Classification (FAC)**	**❌**	**✔**	**✔**	**❌**	**❌**
**Catherine Bergego Scale (CBS)**	**❌**	**✔**	**✔**	**❌**	**✔**
**Lawton and Brody IADL Scale**	**❌**	**✔**	**✔**	**❌**	**❌**

**Table 9 sensors-25-03436-t009:** Accuracy results in detecting patient orientation using Logistic Regression (LR), Decision Tree (DT), Random Forest (RF), Naïve Bayes (NB), K-Nearest Neighbors (KNN), and Support Vector Machine (SVM).

Scale	LR	DT	RF	NB	KNN	SVM
Precision (%)	0.84	0.94	0.94	0.81	0.84	0.91

**Table 10 sensors-25-03436-t010:** Summary of the analyzed parameters, as well as the evaluation criteria for each of them. Where *P*_i_ is the score obtained for each parameter *i*, and the percentage is calculated with respect to the total possible points for each body part.

Body Part	Parameters	Description	Computer Vision Detection	Detection Strategy	Analyzed Motion Ranges	Score by Range	Normalized Score
**Head**	LHTL, LHTR	Lateral head tilt to the left (L) and right (R)	Analysis of the relative position of the head in the frontal body position	Establishing body orientation using AI strategies and calculating torsion angles based on detected keypoints.	0–30° 30–60° 60–90°	[0–25] [25–50] [50–75]	$\sum_{i=1}^{5}\frac{P_{i}}{375}$
	HRL, HRR	Head rotation to the left or right	Analysis of head rotation relative to the body’s axis	Detection of head orientation and rotation using facial reference points and motion analysis.			
	FHT	Frontal head tilt	Detection of forward or backward tilt of the head in relation to the body	Use of facial tracking algorithms to determine the frontal inclination of the head.			
**Arms**	WSAL, WSAR	Wrist–shoulder angle	Detection of joint angles between the wrist and shoulder	Tracking of keypoints in the wrist and shoulder to calculate the angle.	0–60° 60–120° 120–180°	[0–25] [25–50] [50–75]	$\sum_{i=1}^{4}\frac{P_{i}}{300}$
	AELL, AELR	Lateral elevation of extended arms	Detection of lateral elevation of the arms	Tracking of arm trajectory in a lateral plane using limb detection.	0–30° 30–60° 60–90°	[0–25] [25–50] [50–75]	
**Legs**	HAAL, HAAR	Hip–ankle angle	Detection of angles between the hip and ankle	Analysis of leg alignment from the hip to the ankle to calculate the angle.	0–30° 30–60° 60–90°	[0–25] [25–50] [50–75]	$\sum_{i=1}^{4}\frac{P_{i}}{300}$
	LELI, LELR	Lateral elevation of extended legs	Detection of lateral elevation of extended legs	Tracking of leg trajectory in lateral elevation to determine range of motion.			
**Torso**	LTTL, LTTR	Lateral trunk tilt	Detection of lateral tilt of the torso	Use of postural analysis to assess lateral trunk inclination through keypoints in the torso.	0–30° 30–60° 60–90°	[0–25] [25–50] [50–75]	$\sum_{i=1}^{4}\frac{P_{i}}{300}$
	FTIF, FTIB	Frontal trunk inclination	Detection of forward or backward trunk tilt	Tracking of trunk orientation in the frontal plane to determine inclination.			
**Body**	Overall score to establish a global conclusion about the patient’s mobility.						$\sum_{i=1}^{17}\frac{P_{i}}{1575}$

**Table 11 sensors-25-03436-t011:** Mapping of Pose Estimation Parameters to Standardized Clinical Scales for Patient Assessment.

Clinical Scale	Clinical Purpose	Analyzed Information
Motor Index (MI)	Evaluate motor strength and mobility	Wrist–elbow–shoulder angle; Hip–knee–ankle angle; Time in specific positions
Modified Ashworth Scale (MAS)	Measure muscle resistance to passive movement and stiffness	Facial expressions of pain; Wrist–elbow–shoulder and hip–knee–ankle angles (with/without weight); Angular velocity; Movement trajectory
Berg Balance Scale and Shortened Berg Balance Scale	Assess balance and postural control	Head tilt and rotation; Trunk tilt (lateral and frontal); Variations in posture stability
Functional Ambulation Classification (FAC)	Assess independence during gait and limb coordination	Relative distance between keypoints during gait; Movement phases and timing
Catherine Bergego Scale (CBS)	Evaluate neglect and affected limb usage	Asymmetry analysis; Keypoint position monitoring
Barthel Index and/or Lawton and Brody Scale	Assess ability to perform activities of daily living	Body position changes; Time maintaining postures; Stability during tasks
Disability Rating Scale (DRS)	Evaluate disability severity and recovery prognosis	Posture maintenance and activity tracking (keypoints and timing)
National Institutes of Health Stroke Scale (NIHSS)	Evaluate stroke severity	Facial expressions; Arm and leg movement capacity; Trunk stability

## Data Availability

The data supporting the findings of this study are available from the first author upon reasonable request.

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
