# Peer review of "Towards Intelligent Assessment in Personalized Physiotherapy with Computer Vision"

_sensors, 2025, doi:10.3390/s25113436_

Round 1

Reviewer 1 Report

Comments and Suggestions for Authors

The article is very well written, but there are a few things to be considered. I would like to see a detailed comparison of Yolo with other pose estimation models. Here are some additional points to consider:

  1.  If possible, consider at least some preliminary testing with actual physiotherapy patients for better validation 
  2. A bit more quantitative results on the accuracy of clinical scale mapping would be better if the timespan allows you to do so.
  3. Please describe the limitations of the current implementation in detail.

Also, clarify the technical requirements for real-time implementation, and I would like to see more details of real-time implementation, such as the computational cost associated with the Yolo and the variants you use in this study. 

Author Response

Please find attached our response to your comments.

Reviewer 2 Report

Comments and Suggestions for Authors
  1. The article demonstrates how YOLO Pose can efficiently capture human movement without physical sensors, which is beneficial for patient comfort and accessibility.
  2. However, there are major concerns.
  3. The paper provides only preliminary results without thorough validation or benchmarking against existing physiotherapy assessment methods.
  4. There is no mention of testing with actual patients or clinical environments, limiting the evidence of real-world applicability. Page 27, Line 898: "This study would need active involvement of physiotherapy specialists and the collection of real-time patient data." This confirms that the current study has not yet involved clinical data, limiting its real-world applicability. Lack of real patient validation makes the system's performance in clinical environments speculative.
  5. The article does not compare YOLO Pose’s performance with other pose estimation models (OpenPose, MediaPipe), missing an opportunity to justify its choice.
  6. The semantic framework, though conceptually sound, lacks concrete implementation details and validation examples.
  7. The source and nature of data used for evaluating the system (sample size, diversity, type of physiotherapy exercises) are unclear. Page 9, Line 388: “The image dataset was compiled from internet videos focused on rehabilitation movements and mobility exercises.” Using YouTube videos as data sources introduces variability and lacks the rigor of professionally recorded physiotherapy datasets. This reduces the dataset's clinical reliability and standardization.
  8. Metrics such as accuracy, sensitivity, specificity, or computational latency are not reported. Page 27, Lines 893–894: “A critical consideration is the translation of raw pose data into actionable clinical parameters. Using abstraction levels…” The semantic model is still conceptual and not fully validated. No metrics or case studies demonstrate the framework’s effectiveness in mapping sensor data to clinical insight.
  9. The model assumes that all physiotherapy setups have compatible optical sensors, which is not the case, especially in low-resource settings.
  10. There is no discussion on how the system handles body diversity (e.g., different body sizes, mobility levels, or clothing types). 
  11. Page 27, Lines 899–900: “...facial expression analysis, which is not considered in this work, should be included to develop a more general semantic framework…” Pain and emotional state are crucial in physiotherapy assessment. Absence of this consideration results in an incomplete patient evaluation model.
  12. Capturing patient video data raises privacy and ethical issues that are not addressed in the manuscript. Page 6, Lines 266–273: “Privacy and data confidentiality are crucial... must implement robust security protocols…” Although risks are noted, the study does not implement or evaluate any privacy-preserving strategies. Ethical issues remain unaddressed at the implementation level.
  13. The paper does not discuss how the system would scale to large clinical environments or home-based physiotherapy settings.
  14. No discussion is provided on common pose estimation errors or strategies to mitigate them.
  15. The mention of NLP support is superficial and lacks depth or justification.
  16. There is no consideration of how clinicians or patients interact with the system (user interface design or ease of use).
  17. While the paper argues for better assessment, it does not show how its outputs would concretely improve or alter clinical decisions or outcomes.

Author Response

(The authors gave the same response as above.)

Round 2

Reviewer 1 Report

Comments and Suggestions for Authors

The author's responses to the comments are good enough. 

Reviewer 2 Report

Comments and Suggestions for Authors

I am satisfied with the author response and the revised manuscript.